# Influence of Extremely Low Temperatures of the Pole of Cold on the Lipid and Fatty-Acid Composition of Aerial Parts of the Horsetail Family (Equisetaceae)

**DOI:** 10.3390/plants10050996

**Published:** 2021-05-17

**Authors:** Vasiliy V. Nokhsorov, Lyubov V. Dudareva, Svetlana V. Senik, Nadezhda K. Chirikova, Klim A. Petrov

**Affiliations:** 1Institute of Natural Sciences, North-Eastern Federal University, 48 Kulakovskogo Str., 677000 Yakutsk, Russia; hofnung@mail.ru; 2Laboratory of Physical and Chemical Research Methods, Siberian Institute of Plant Physiology and Biochemistry, Siberian Branch of Russian Academy of Sciences, 132 Lermontova Str., 664033 Irkutsk, Russia; laser@sifibr.irk.ru; 3Laboratory of Fungal Biochemistry, Komarov Botanical Institute, Russian Academy of Sciences, 2 Professor Popov Str., 197376 St. Petersburg, Russia; senik@binran.ru; 4Laboratory of Biogeochemical Cycles of Permafrost Ecosystems, Institute for Biological Problems of Cryolithozone of Siberian Branch of the Russian Academy of Sciences, 41 Lenina Av., 677000 Yakutsk, Russia; kap_75@bk.ru

**Keywords:** *Equisetum variegatum*, *Equisetum scirpoides*, horsetails, low temperatures, hardening, phospholipids, glycolipids, betaine lipids, DGTS, fatty acids, seasonal dynamics, the Pole of Cold

## Abstract

The lipid composition of two species of vascular plants, *Equisetum variegatum* Schleich. ex. Web. and *E. scirpoides* Michx., growing in the permafrost zone (Northeastern Yakutia, the Pole of Cold of the Northern Hemisphere), with average daily air temperatures in summer of +17.8 °C, in autumn of +0.6 °C, and in winter of −46.7 °C, was comparatively studied. The most significant seasonal trend of lipid composition was an accumulation of PA in both horsetail species in the autumn–winter period. Cold acclimation in autumn was accompanied by a decrease in the proportion of bilayer-forming lipids (phosphatidylcholine in the non-photosynthetic membranes and MGDG in photosynthetic membranes), an increase in the desaturation degree due to the accumulation of triene fatty acids (*E. scirpoides*), and an accumulation of betaine lipids *O*-(1,2-diacylglycero)-*N*,*N*,*N*-trimethylhomoserine (DGTS). The inverse changes in some parameters were registered in the winter period, including an increase in the proportion of “bilayer” lipids and decrease in the unsaturation degree. According to the data obtained, it can be concluded that high levels of accumulation of membrane lipids and polyunsaturated FAs (PUFAs), as well as the presence of Δ5 FAs in lipids, are apparently features of cold hardening of perennial herbaceous plants in the cryolithozone.

## 1. Introduction

The study of organisms adapting to extreme environmental conditions is one of the important problems of modern biology. Northeastern Yakutia is a physico-geographical region with severe and very diverse living conditions. The Pole of Cold of the Northern Hemisphere is located here. The entire region lies in the permafrost zone, the thickness of which exceeds 600 m in some places. Many plant species inhabiting Northeastern Yakutia grow there at their distribution limit.

The specificity of the seasonal growth of the bulk of the herbaceous vegetation in Northeastern Yakutia is that it develops intensively in the first half of summer in order to have time to go through a complete vegetation cycle and give full-fledged seeds. However, horsetail meadows, especially those consisting of variegated horsetail (*E. variegatum*) (Figure 1), are often subjected to long-term hardening by floodwaters. Under these conditions, they do not have time to go through the complete vegetation cycle. When the winter season begins, autumn-vegetating plants get covered with snow with their shoots green and acquire the ability to accumulate relatively more storage lipids and other nutrients [1,2].

At first glance, it seems incredible that herbaceous plants at the Pole of Cold manage to enrich themselves with a large number of the most energy- and material-intensive substances during the growing season. Nevertheless, horsetails accumulate a large amount of proteins, carbohydrates, and lipids, and they become an adaptive fattening food for both non-hibernating and hibernating herbivores in autumn, winter, and early spring [3,4]. This rapid accumulation of nutrients is supported by a complex of evolutionary adaptations to permafrost conditions. For example, we previously determined that *E. variegatum* (like *E. arvense* and *E. scirpoides*) belongs to the group of 16:3 plants in terms of the content of triene FAs. Specifically, Δ5 acids, including 5,11,14,17-eicosatetraenoic acid (uniperonic), which are characteristic of evolutionarily ancient taxa, were identified in the FA composition of the aerial parts of horsetails. Four unsaturated bonds in this fatty acid are assumed to be possibly involved in the enhancement of plant tissue resistance to low temperatures. [5]. It was also found that, in the autumn, when adapting to low ambient temperatures of the Pole of Cold in the Northern Hemisphere, horsetails synthesize a secondary keto carotenoid—rhodoxanthin. It was shown that secondary carotenoids can play a special role in the regulation of tolerance to low temperatures in some plant species (for instance, *Equisetum*). Thus, rhodoxanthin in some cases features a more important antioxidant function than the xanthophyll cycle, which cannot operate at constantly low temperatures [6].

Low positive temperatures (0 + 10 °C), named a cold spell, lead to acclimation (hardening). It is an adaptation response during which plants trigger numerous biochemical and physiological mechanisms necessary for acquiring frost resistance and survival in winter [7,8]. Subzero temperatures (frosts) cause injuries in plant tissues due to the formation of ice crystals. The plasma membrane is considered to be the primary site of freezing injury and, therefore, is subjected to restructuring during cold acclimation. According to modern concepts, the relative abundance and saturation level of membrane lipids play an important role in the process of plant adaptation to temperature stress. Phosphatidylcholine (PC) and phosphatidylethanolamines (PEs) are the main lipids of plasma membranes, mitochondrial membranes, and microsomes. Adaptation to low temperature is accompanied by an increase in the proportion of lipids that destabilize the bilayer membrane, such as PE and phosphatidic acid (PA) [9], and by an increase in the level of fatty-acid unsaturation that increases the phase separation of lipids [10,11,12]. This restructuring of the lipid composition allows membranes to remain fluid. This is necessary for normal growth. Less is known about cold adaptation of chloroplast membranes consisting mainly of galactolipids, sulfolipids, and phosphatidylglycerol (PG). MGDGs are known to be localized in thylakoid membranes and are associated with photosystems I and II. DGDGs are the main lipids of both the thylakoid membranes and the outer shell of chloroplasts. They play an important role in the membrane structure and stabilization of proteins in the membrane. DGDGs are required in large amounts during the period of active growth for normal plant development. SQDGs are located mainly in chloroplast lamellae [13]. PG is known to be involved in the stabilization of PS II and light-harvesting complex II.

The purpose of this paper was a comparative analysis of the seasonal (summer, autumn, and winter) dynamics of the content of membrane lipids and the FA composition of total lipids in the aerial parts of *E. variegatum* and *E. scirpoides* growing in the cryolithozone of Northeastern Yakutia (the Pole of Cold).

## 2. Results

### 2.1. Lipid Content

Figure 2 and Figure 3 show the seasonal variability of polar lipids in the aerial parts of *E. variegatum* and *E. scirpoides*. The content of phospholipids (PLs) was characterized by individual seasonal dynamics. Phosphatidic acid (PA) in *E. variegatum* and phosphatidylcholine (PC) in *E. scirpoides* were the main PLs of the plasma membrane. Both plants accumulated PA in response to the decrease in air temperature and photoperiod in the autumn–winter period. Other lipids demonstrated different seasonal trends in the two plants studied. Phosphatidylcholine (PC) was accumulated in *E. variegatum* in winter when the ambient temperature was extremely low (−46.7 °C), whereas, in *E. scirpoides*, this lipid content increased in autumn but decreased in winter. The content of phosphatidylethanolamines (PEs) decreased in both plants in winter as compared to the autumn value, but summer values of this lipid were species-specific.

Lipid content trends become more understandable if we consider the ratios of different groups of lipids. During adaptation to temperature shifts, membranes maintain a balance between “bilayer” and “non-bilayer” lipids [9]. The ratio of “bilayer” (PC) and “non-bilayer” (PE and PA) lipids in *E. variegatum* was 0.4 in summer, before decreasing to 0.25 in autumn, and then it was restored to 0.45 in winter. A similar trend was observed in *E. scirpoides*; the PC/(PE + PA) value was 1.4 in summer, decreased to 1.0 in autumn, and was restored to 1.2 in winter.

The level of phosphatidylglycerols (PG) in *E. variegatum* increased in the autumn–winter period. On the contrary, in *E. scirpoides*, the PG content sharply decreased from summer to the autumn–winter period. The content of DPGs (diphosphatidylglycerols) in *E. variegatum* gradually increased in autumn. Opposite changes in this lipid were observed in *E. scirpoides*.

The content of betaine lipid DGTS in both horsetail species increased in the cold season. In *E. variegatum*, the DGTS accumulation in shoots was noted earlier (just after the onset of low positive hardening temperatures) to a much greater extent (8.8-fold, up to 1.7 mg/g dry weight in September), while, in *E. scirpoides*, a twofold increase in the DGTS content was observed only after the onset of extremely low temperatures in November and reached only 0.9 mg/g dry weight.

After the end of the horsetail summer growing season and with a sharp break of low positive and extremely low negative air temperatures, the content of glycolipids (GLs) in the studied species changed differently. The content of monogalactosyldiglycerides (MGDGs) in *E. scirpoides* was maximal in summer and decreased sharply in the autumn–winter period (3.9-fold and 5.4-fold, respectively). In *E. variegatum*, the MGDG level decreased 2.1-fold and was restored to the same value in winter. The level of DGDG in *E. variegatum* was higher in the autumn–winter period as compared with summer. In *E. scirpoides*, on the contrary, the DGDG content decreased twofold in winter. We suppose that a certain role in cold acclimation is played by the MGDG/DGDG ratio. Its highest value was observed in summer in both plants (1.3 in *E. variegatum* and 0.7 in *E. scirpoides*). It then decreased in autumn (0.4 in *E. variegatum* and 0.2 in *E. scirpoides*) and was 0.5 in winter in both plants.

The content of sulfoquinovosyldiacylglycerols (SQDGs) in *E. scirpoides* decreased twofold in autumn compared to the summer values. *E. variegatum* demonstrated the maximal SQDG level in autumn, when it doubled the summer values.

Glycoceramides (GlCers) were found in the content of sphingolipids. In winter, *E. variegatum* showed a twofold increase in the GlCer content in comparison with the summer and autumn periods. The GlCer content in *E. scirpoides* remained practically unchanged during temperature acclimation.

### 2.2. Fatty-Acid Composition

The analysis of seasonal changes in the FA composition of horsetail shoots revealed a rather wide variety of both saturated and unsaturated mono-, di-, tri-, and tetraene FAs with a *cis*-configuration of double bonds (Table 1 and Table 2). In the FA profile of *E. variegatum*, 18 and 15 FAs were identified during summer and autumn, respectively; 16 FAs were identified in the winter period. Among saturated FAs, palmitic acid C16:0 prevailed in all seasons. In the autumn–winter period, its content was 4–6% higher than in the summer period.

In another representative of fattening horsetails, *E. scirpoides*, the qualitative composition of lipid FAs in summer and autumn vegetative shoots was also quite different from winter plants of the same species (Table 2). Thus, 19 individual FAs were identified in summer and autumn; 15 FAs were identified in winter. *E. scirpoides* plants that were covered with snow in the frozen state differed significantly from summer non-vegetating plants in terms of the absolute FA content level. Thus, the total FA content increased by 4.6 mg/g in autumn and by 6.3 mg/g dry weight in winter, as compared to the summer values. Acids with an odd number of carbon atoms (C15:0 and C17:0) were found in small quantities in both horsetail species. Monounsaturated FAs with 16, 18, and 20 carbon atoms were also found in the species studied. The relative content of palmitoleic C16:1 acid in summer-vegetating *E. variegatum* was more than twofold higher than that in autumn and winter plants. In general, in the species studied, the content of monoenoic acids was significantly lower than the content of polyeneic ones and did not exceed 8.4% of the total FA content.

PUFAs in horsetail tissues are represented by Δ12, Δ15, Δ5 diene, triene, and tetraene acids. Linoleic acid C18:2 was the major diene FA in all seasons. In *E. variegatum*, the content of this acid in autumn reached 23.3% ± 1.5% of the total FA, which is 12% higher than in summer and 8.1% higher than in winter. In *E. scirpoides*, the C18:2 content in the summer–autumn period varied within narrow limits: 2–2.3 mg/g dry weight. In winter, its content increased to 18% of the total FA. A small amount of hexadecadienoic acid C16:2 (Δ7.10), which is a precursor in the biosynthesis of hexadecatrienoic acid C16:3 (Δ7,10,13), was identified in the tissues of the studied plants. Plants containing a significant amount of C16:3 (Δ7,10,13) in lipids of photosynthetic tissues, mainly in the MGDG composition, belong to the 16:3 type [12]. Hexadecatrienoic acid was found in all seasons in both species studied. *E. variegatum* had a higher content of C16:3 (Δ7,10,13) in summer (6.2%) and winter (4%) and a lower content in autumn (1.3%). That coincides with the seasonal dynamics in MGDG content. In *E. scirpoides*, during the entire autumn–winter period, the content of C16:3 (Δ7,10,13) was at a consistently high level (5–6%) despite the depletion of MGDG.

In all seasons, the main PUFA in the two studied species was α-linolenic acid. In *E. variegatum*, its content was maximum (6.6 mg/g dry weight) in the winter period (22 November). In *E. scirpoides*, its content was maximum (6.5 mg/g dry weight) in autumn (25 September).

In the tissues of *E. scirpoides*, Δ5 FAs were found: C18:2 (Δ5.9) taxoleic acid and C20:4 (Δ5,11,14,17) uniperonic acid. In *E. variegatum*, only uniperonic acid was found. With the onset of winter extreme air temperatures, the content of uniperonic acid in *E. variegatum* increased to 3% of the total FA, which was 0.6 mg/g dry weight. The content of this acid in *E. scirpoides* doubled (up to 0.6 mg/g dry weight) with the onset of autumn hardening cold weather, snow, and a decrease in the length of daylight hours. With the onset of winter frosts, i.e., critically low air temperatures characteristic of the Pole of Cold in the Northern Hemisphere, the content of uniperonic acid in *E. scirpoides* decreased to 1.7% of the total FA. In the studied species, C20:3 acid (Δ11,14,17), a necessary precursor in the biosynthesis of C20:4 (Δ5,11,14,17) uniperonic acid, was identified [14]. The relative content of C20:3 (Δ11,14,17) in both types of horsetails increased with the break of unfavorable living conditions in the autumn period. The most pronounced increase in this acid content was noted in *E. variegatum*: 3.1-fold higher, compared with the summer values. In *E. scirpoides*, it was 0.9-fold higher. Seasonal changes in the absolute and relative FA content of total lipids and the calculated activities of the corresponding desaturases were reflected in the changes in the desaturation ratios: stearoyl desaturation ratio (SDR), oleoyl desaturation ratio (ODR), and linoleoyl desaturation ratio (LDR).

## 3. Discussion

Having analyzed the data obtained, we can note certain trends in changes in the lipid content and their fatty-acid composition in the studied horsetail species taken for analysis during the growing season (in summer, autumn, and winter) at different temperatures of the cryolithozone. All the evidence now suggests that the sharp continental climate contributed to the fact that, in the process of long evolution, vascular plants of permafrost ecosystems developed complex mechanisms of biochemical adaptation to extremely low temperatures. Among the mechanisms, lipid metabolism and their special FA composition play a significant role.

The adaptation to cold stress is well documented to be accompanied by an increase in the proportion of “non-bilayer” lipids that destabilize the bilayer membrane to maintain its fluid state under low temperature conditions [9,15]. According to the data obtained, the ratios of “bilayer“ and “non-bilayer“ lipids were specific for each of the two studied species and varied in the range of 0.25–4.5 for *E. variegatum* and 1.0–1.4 for *E. scirpoides*. However, the dynamics of this ratio during cold acclimation of the two horsetails was the same: decrease in “bilayer“ lipids in September during hardening with low positive autumn temperatures (+0.6 °C) and restoration of the summer values in winter with extremely low temperatures (−46.7 °C).

A similar trend was registered for lipids of photosynthetic membranes. The main lipids of chloroplast membranes are MGDG, DGDG, SQDG, and PG. Having a galactose headgroup and two highly unsaturated fatty acid chains, MGDG has a cone shape and tends to form non-bilayer membrane structures [16]. DGDG, SQDG, and PG have a cylindrical shape and form bilayers [17]. A high proportion of MGDG in chloroplast membranes stabilizes membrane regions with concave curvature, promotes the formation of thylakoid stacks [18], and is important for the chloroplast shape [19]. Therefore, accurate tuning of the MGDG/DGDG ratio is thought to be important for the optimization of photosynthesis in environmental conditions, for example, salinity stress [20]. Similar to the ratio of “bilayer” and “non-bilayer” lipids of non-photosynthetic membranes, the MGDG/DGDG ratio in the two horsetails studied was higher in summer, before decreasing in autumn, and it was partly restored in winter. The ratio of MGDG to all “non-bilayer” chloroplast lipids (DGDG + SQDG + PG) demonstrated the same seasonal dynamics. A reduction in the MGDG level under frost stress was registered in *Arabidopsis* [21]. The authors supposed that it facilitated the maintenance of the lamellar chloroplast outer envelope membrane and prevented the outer membrane from fusing with other cellular membranes during frost stress. Another explanation for the MGDG decrease in autumn–winter period is that these lipids are associated with photosystems I and II; thus, their content may be correlated with the photosynthetic activity of horsetails. During a short growing season at high air temperatures, horsetails are known to grow actively and accumulate biomass. Most of them mature spores. During this period, the processes of photosynthesis are intensified, and there is a need for a large amount of MGDG. The MGDG content decreases in September with the onset of autumn cold weather and a decrease in the photoperiod in this season.

Another significant trend was represented by the accumulation of PA in the autumn–winter period in both horsetail species. PA is known to be an important signaling lipid in plant responses to abiotic stress [15]. Low, nonfreezing temperatures were found to trigger fast and sustained PA increase in *Arabidopsis thaliana* through the phosphorylation of DAG by diacylglycerol kinase [22] or through the hydrolysis of phospholipids (predominantly PC) by phospholipase D [23]. According to the data presented in Figure 3, PA accumulation in *E. scirpoides* occurred in parallel with the sharp decrease in PC and was most likely caused by the action of phospholipase D. The high PA level in optimal temperature conditions in *E. variegatum* may be the evolutionary developed adaptation of this horsetail to permafrost conditions.

The presence of betaine lipids (BL), in particular *O*-(1,2-diacylglycero)-*N*,*N*,*N*-trimethylhomoserine (DGTS), in the lipid profile of both horsetail species and its accumulation in the autumn–winter period are of particular interest. DGTSs are believed to be an evolutionarily ancient class of lipids and are synthesized by some unicellular seaweeds, higher vascular plants such as horsetails, ferns, and mosses, some fungi, and protozoan tissues [24,25,26,27]. Under the influence of various types of stress (osmotic, water, phosphorus deficiency, etc.) the DGTS content can increase [24]. This is considered as one of the ancient mechanisms of biochemical adaptation of plants with the participation of lipid molecules [28]. The authors of [25] emphasized that DGTS plays the same role in the cells of lower plants as PC in higher plants. In the autumn period, the DGTS content in *E. variegatum* was significantly higher than the PC content. DGTS can possibly functionally replace PC in *E. variegatum* tissues since the DGTS chemical structure is close to the structure of this most widespread phospholipid [24,27]. In *E. scirpoides*, however, the high PC content in its tissues during all vegetation periods did not apparently need compensation by DGTS. The BL content in this species remained low (0.6 mg/g dry weight in the winter period). In many cases, as reported in the literature, BL is not synthesized under optimal cultivation conditions and accumulates under stress conditions. Whereas the activation of the DGTS biosynthesis in response to phosphorus starvation is well studied [24,26,27,29], less is known about the involvement of DGTS in plant adaptation to cold. However, it was reported that the amount of this lipid varies in different parts of ferns (clefs and fronds) depending on the season, with the minimum in fronds in mid-summer [30]. It is possible that changes in the DGTS content in the horsetails are caused not so much by air temperature as by the growth stage.

The functional properties and state of membranes largely depend on the lipid bilayer FA composition. The analysis of the FA composition dynamics of horsetail lipids during hardening by low positive autumn temperatures revealed an increase in the absolute amounts of unsaturated FAs (Table 1 and Table 2). The degree of total lipid unsaturation, expressed as the coefficient k, increased only in *E. scirpoides*. The adaptive changes in the FA composition of this horsetail occurred due to an increase in the proportion of triene fatty acids (hexadecadienoic C16:3 (Δ7,10,13) and α-linolenic C18:3 (n-3)). The amount of unsaturated FAs is known to significantly affect the membrane permeability and the activity of many membrane-bound enzymes [31]. A high degree of FA unsaturation is believed to determine the ability of organisms to adapt to low temperatures due to the less dense packing of molecules in the membrane bilayer. Unexpectedly, the degree of the total lipid unsaturation decreased in winter with extremely low temperatures in both species studied.

In horsetail lipids, we identified Δ5 FAs. As a rule, they are present in lipids of gymnosperms in varying amounts. Their physiological role is poorly understood, but it is assumed that their presence in lipids of gymnosperms is associated with plant resistance to low ambient temperatures. For instance, this was confirmed by the absence of Δ5 FAs in the leaf lipids of *Welwitschia mirabilis,* an inhabitant of tropical water bodies [32], and their high content in frost-hardy conifers [33]. The Δ5 FAs are found in lipid compositions of other low temperature-tolerant organisms, for example, in some lichens [34]. The results of the analysis of changes in the content of Δ5 acids in the tissues of horsetails during the growing season did not allow us to draw an unambiguous conclusion about their participation in the formation of the resistance to hypothermia in these horsetail species.

Thus, some of the identified trends in lipid seasonal dynamics were common to both horsetails studied, while others were species-specific. The most significant trend in the autumn–winter period was the accumulation of PA in both horsetail species. Cold acclimation in autumn was shown to be accompanied by a decrease in the proportion of “bilayer” lipids of plasma and photosynthetic membranes and an accumulation of betaine lipids (both horsetails) or an increase in the desaturation degree due to the accumulation of triene fatty acids (*E. scirpoides*). It is surprising that the inverse changes in some parameters were registered in winter, including an increase in the proportion of “bilayer” lipids and a decrease in the unsaturation degree. Similar changes occur during deacclimation, a process describing the reduction in frost resistance that was originally attained during acclimation [35]. The molecular basis of this process is still largely unknown, but it is obviously related to the exposure to warmer temperatures in spring, as well as the altered photoperiod and dormancy status of plants. At the Pole of Cold, horsetails overwinter under a deep (1–1.5 m) layer of snow. The ambient temperature under snow is known to be much higher than in the open air and is not as susceptible to sudden changes as in snow-free autumn. These features of the temperature regime possibly contribute to the seasonal dynamics of lipids described in this paper.

## 4. Materials and Methods

The objects of the study were the aerial parts of variegated horsetail (*E. variegatum*) and reed horsetail (*E. scirpoides*) (Figure 1). Equisetaceae plant samples were collected as an average biomass (2–5 g) consisting of several identical plants without a root system from a 1 m^2^ plot in triplicate. The samples were immediately fixed in liquid nitrogen and transported in Dewar vessels to the laboratory. For biochemical studies, the samples of the aerial parts of horsetails fixed in liquid nitrogen were dried in the lyophilizer (VirTis, New York, USA). The time and the place of sampling were as follows: summer vegetation—early August; autumn vegetation—late September; winter vegetation—late November, Northeastern Yakutia, 67° N, 137° E (the Pole Cold of the Northern Hemisphere). Data on air temperature at the habitat of perennial herbaceous plants (the Pole Cold) were taken from the Internet resource “Weather Underground” (http://www.wunderground.com (accessed on 17 September 2020)). Weather conditions during the years of the experiment were typical for Northeastern Yakutia. The onset of frost was noted in the middle of the second decade of September, and a steady transition of the night temperature through 0 °C was noted in late September and early October (Figure 4).

For lipid extraction, a weighed portion of plant material (0.5 g) was fixed in liquid nitrogen and ground until a homogeneous mass was obtained. Cooled laboratory glassware and reagents were used. Then, 10 mL of a 1:2 chloroform/methanol mixture was added. Ionol was added to the mixture as an antioxidant (0.00125 g per 100 mL of the mixture). Everything was thoroughly mixed and left for 30 min until the complete diffusion of lipids into the solvent. The solution was transferred quantitatively into a separatory funnel through a filter. The mortar and filter were washed three times with the same solvent mixture. To separate the nonlipid components, water was added.

For the analysis of total lipids, the lower chloroform fraction was separated. Chloroform (high purity grade, stabilized with 0.005% amylene) was removed from the lipid extract under vacuum using the RVO-64 rotary evaporator (Mikrotechna, Praha, Czech Republic). To control the lipid extractability (%), the known amount of 10 µg of nonadecanoic acid (C19:0) was added at the homogenization stage. FA methyl esters (FAMEs) were obtained using the Christie method [36]. Additional FAME purification was carried out by thin-layer chromatography (TLC) on glass plates with KSK silica gel (Reachem, Moscow, Russia). Benzene was used as the mobile phase. To visualize the FAME zone (Rf = 0.71–0.73), the plates were sprayed with 10% H_2_SO_4_ in MeOH and heated in an oven at 100 °C. The FAME zone was removed from the plate with a spatula and eluted from silica gel with *n*-hexane. The FAME analysis was performed by GLC using the 5973/6890N MSD/DS gas chromatograph–mass spectrometer (Agilent Technologies, Santa Clara, CA, USA). The detector was a quadrupole mass spectrometer. The ionization method was electron impact with an ionization energy of 70 eV. The analysis was performed in the mode of the total ion current recording.

An HP-INNOWAX capillary column (30 m × 250 μm × 0.50 μm) with a stationary phase (PEG) was used to separate the FAME mixture. The carrier gas was helium, and the gas flow rate was 1 mL/min. The evaporator temperature was 250 °C, the ion source temperature was 230 °C, and the detector temperature was 150 °C. The temperature of the line connecting the chromatograph with the mass spectrometer was 280 °C. The scanning range was 41–450 amu. The volume of the injected sample was 1 μL, and the flow divider was 5:1. The separation of the FAME mixture was carried out in isothermal mode at 200 °C. To identify FAs, the NIST 08 mass spectral library and the Christie FAME mass spectral archive were used [37]. The relative FA content was determined by the method of internal normalization in weight percent (wt.%) of the total content in the test sample, taking into account the FA response coefficient.

To characterize the degree of lipid unsaturation, the unsaturation coefficient (*k*) and the double-bond index (DBI) were used [38].
*k* = ∑Punsaturated/∑Psaturated,DBI = ∑Pj nj/100,
where P is the acid content (%), Pj is the acid content (%), and nj is the number of double bonds in each acid.

The activity of acyl-lipid membrane ω9-, ω6-, and ω3-desaturases responsible for the introduction of double bonds into hydrocarbon chains of oleic (C18:1 (n-9)), linoleic (C18:2 (n-6)), and α-linolenic (C18:3 (n-3)) fatty acids was calculated as stearoyl (SDR), oleyl (ODR), and linoleyl (LDR) desaturase ratios [39] formulas as follows:

SDR = (%C18:1)/(%C18:0 + %C18:1),
ODR = (%C18:2 + %C18:3)/(%C18:1 + %C18:2 + %C18:3),
LDR = (%C18:3)/(%C18:2 + %C18:3).

Individual classes of polar lipids were analyzed by two-dimensional TLC on silica gel plates 60 (10 × 10 cm) (Merck, Darmstadt, Germany) in a system of solvents: chloroform–methanol–water (65:25:4) in the first case and chloroform–acetone–methanol–acetic acid–water (50:20:10:10:5) in the second case [27].

The lipids were identified using standards for target components and specific reagents for individual functional groups [40].

The amounts of phospho-, glyco-, sphingo-, and betaine lipids were determined densitometrically using the Denskan (Lenkhrom, St. Petersburg, Russia). For this purpose, chromatograms were developed in 10% sulfuric acid in methanol followed by heating at 140 °C. The calculation of the content of individual classes of lipids in chromatograms was carried out using the DENS-14-12-03 program in linear approximation mode on calibration curves constructed using standard PC solutions (Larodan, Solna, Sweden), bovine cerebrosides, and MGDG (Sigma, St. Louis, MO, USA).

*Statistical processing*. The tables show the average data from three biological replicates and their standard deviations. The experimental data were statistically processed using the statistical analysis package in the Microsoft Office Excel 2017 environment. The statistical significance of the differences between the compared mean values was assessed using the *t*-test (*p* < 0.05).

## 5. Conclusions

*E. scirpoides* and *E. variegatum*, circumpolar hypoarctic species growing at the Pole of Cold, are of great importance as the valuable autumn–winter fattening food for northern mammals. The main stages of lipid synthesis in the cells of these plants are associated with the formation of glycerol and PUFAs from sugars. The process of accumulation of carbohydrates is most pronounced during the period of cold hardening of plants by low positive temperatures [3,11,41]. According to the data obtained, increases in the total lipid content and the unsaturation level are also observed in the autumn–winter period. Autumn vegetative plants, called cryofood, enter the body of animals, where PUFAs are formed from sugars [4]. Upon receiving C18:2 (n-6) linoleic acid and C18:3 (n-3) α-linolenic acid with food, animals are able to synthesize physiologically valuable long-chain arachidonic PUFAs C20:4 (n-6), eicosapentaenoic C20:5 (n-3), and docosahexaenoic C22:6 (n-3) acids [42].

The problem of breeding new frost-resistant high-yielding varieties of perennial grasses in the permafrost zone of Yakutia is quite difficult because the varieties have to be modeled adapted to extremely severe natural conditions and to an almost intensive year-round use of herbage on permafrost soils. Compared to other regions of the world, there is no long grass breeding experience in the permafrost zone of Yakutia. The data obtained expand the modern understanding of the involvement of lipids in the formation of cold and frost resistance of plants under the action of abiotic factors in the extreme climatic conditions of the cryolithozone. The factual findings on the accumulation of lipid components in the tissues of the horsetails during cold hardening are of great practical importance since they indicate the possibility of the high nutritional value formation of autumn-vegetating plants and winter-green herbaceous ones frozen by natural cold. Such cryofood provides energy for the vital functions of the domestic and wild animals of the North that feed on it.

In general, on the basis of the abovementioned results, the authors came to the conclusion that the lipid and fatty-acid composition of autumn-vegetating and winter-green plants of Northeastern Yakutia play a key role in the regulation of the vital activity of herbivores and humans through the following food chain: cryofood → animals → humans.

## Figures and Tables

**Figure 1 plants-10-00996-f001:**
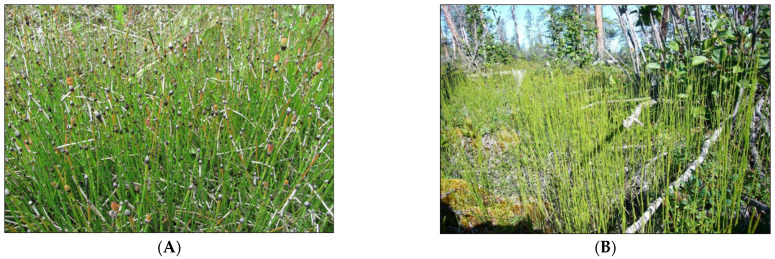
Summer-growing shoots of *Equisetum variegatum* (**A**) and *Equisetum scirpoides* (**B**) growing at the Pole of Cold of the Northern Hemisphere.

**Figure 2 plants-10-00996-f002:**
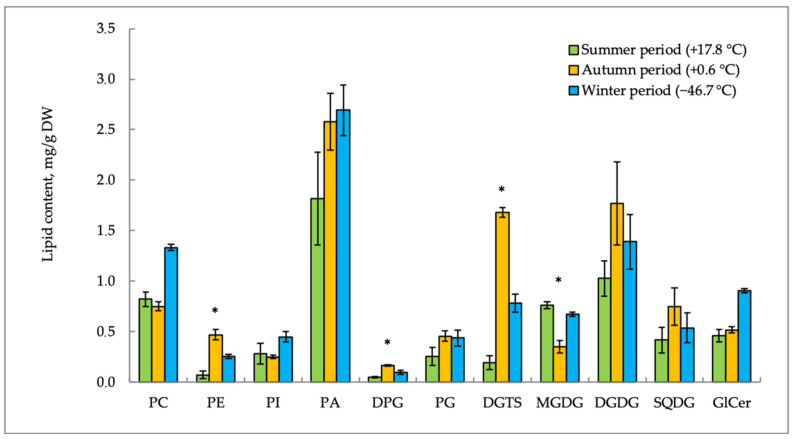
Seasonal changes in the content of individual lipid classes in *E. variegatum* shoots. PC—phosphatidylcholine; PI—phosphatidylinositol; PE—phosphatidylethanolamine; PG—phosphatidylglycerol; PA—phosphatidic acid; DPG—diphosphatidylglycerol; GlCer—glycoceramide, DGDG—digalactosyldiglyceride, DGTS—*O*-(1,2-diacylglycero)-*N*,*N*,*N*-trimethylhomoserine, MGDG—monogalactosyldiglyceride, SQDG—sulfoquinovosyldiacylglycerol. Data are shown as the mean ± SD (*n* = 3); * significant differences according to Student’s *t*-test.

**Figure 3 plants-10-00996-f003:**
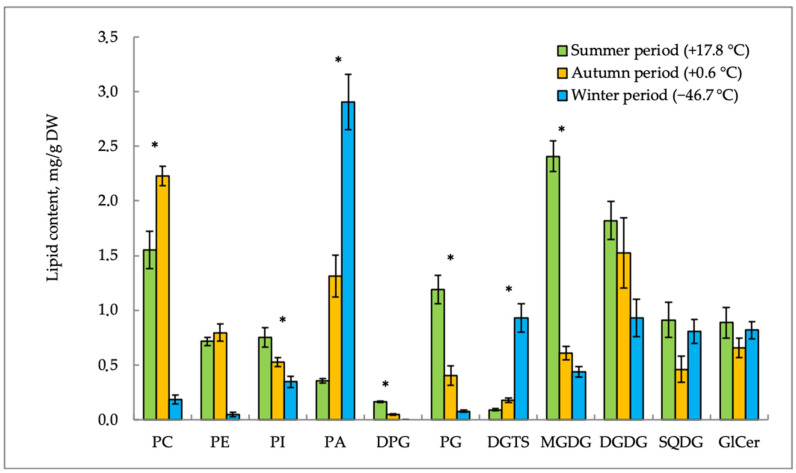
Seasonal changes in the content of individual lipid classes in *E. scirpoides* shoots. PC—phosphatidylcholine; PI—phosphatidylinositol; PE—phosphatidylethanolamine; PG—phosphatidylglycerol; PA—phosphatidic acid; DPG—diphosphatidylglycerol; GlCer—glycoceramide, DGDG—digalactosyldiglyceride, DGTS—*O*-(1,2-diacylglycero)-*N*,*N*,*N*-trimethylhomoserine, MGDG—monogalactosyldiglyceride, SQDG—sulfoquinovosyldiacylglycerol. Data are shown as the mean ± SD (*n* = 3); * significant differences according to Student’s *t*-test.

**Figure 4 plants-10-00996-f004:**
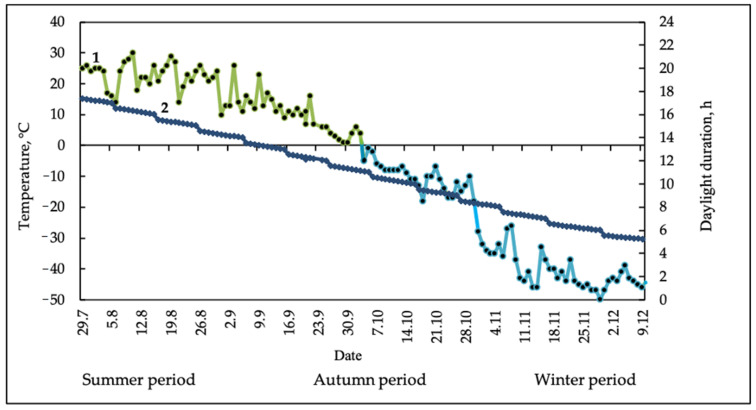
Seasonal changes in air temperature (1) (green line—positive temperatures, blue line—negative temperatures) and daylight duration (2) in Northeastern Yakutia (the Pole of Cold the Northern Hemisphere) in 2019. Air temperature is represented by mean daily values.

**Table 1 plants-10-00996-t001:** Seasonal changes in the FA composition of lipids in shoots of *Equisetum variegatum* growing at the Pole of Cold.

Fatty Acids	Summer Period (+17.8 °C *)		Autumn Period (+0.6 °C *)		Winter Period (−46.7 °C *)	
	mg/g DW	%	mg/g DW	%	mg/g DW	%
C12:0	traces	0.2 ± 0.0 ^a^	–	–	–	–
C14:0	0.1 ± 0.0 ^a^	1.2 ± 0.4 ^a^	0.1 ± 0.0 ^b^	0.9 ± 0.1 ^a^	0.2 ± 0.0 ^b^	0.9 ± 0.2 ^a^
C15:0	traces	0.4 ± 0.0 ^a^	0.1 ± 0.0 ^a^	0.5 ± 0.2 ^a^	traces	0.2 ± 0.0 ^b^
C16:0	2.9 ± 0.7 ^a^	24.7 ± 1.1 ^a^	4.2 ± 0.7 ^a^	26.7 ± 2.1 ^c^	6.5 ± 1.2 ^b^	30.6 ± 1.3 ^c^
C16:1	0.3 ± 0.1 ^a^	2.4 ± 0.8 ^b^	0.2 ± 0.0 ^a^	1 ± 0.2 ^a^	0.1 ± 0.0 ^a^	0.6 ± 0.1 ^a^
C16:2(n-6)	0.1 ± 0.0 ^b^	1 ± 0.3 ^a^	0.1 ± 0.0 ^b^	0.7 ± 0.1 ^b^	0.1 ± 0.0 ^a^	0.7 ± 0.1 ^a^
C16:3(Δ7,11,14)	0.7 ± 0.2 ^a^	6.2 ± 1.2 ^b^	0.2 ± 0.0 ^a^	1.3 ± 0.3 ^a^	0.8 ± 0.1 ^a^	4 ± 0.9 ^b^
C17:0	traces	0.3 ± 0.0	0.1 ± 0.0 ^b^	0.6 ± 0.1 ^a^	0.1 ± 0.0 ^b^	0.5 ± 0.1 ^a^
C18:0	0.2 ± 0.0 ^b^	2.1 ± 0.9 ^a^	0.5 ± 0.1 ^a^	3.3 ± 0.8 ^b^	0.5 ± 0.1 ^a^	2.3 ± 0.8 ^b^
C18:1(n-9)	0.6 ± 0.1 ^a^	5.2 ± 1.2 ^b^	0.9 ± 0.3 ^a^	5.5 ± 1.4 ^c^	1 ± 0.2 ^a^	4.6 ± 0.9 ^c^
C18:1(n-7)	0.1 ± 0.0 ^b^	0.5 ± 0.1 ^a^	–	–	0.1 ± 0.0 ^a^	0.5 ± 0.0 ^a^
C18:2(n-6)	1.3 ± 0.3 ^b^	11.3 ± 1.2 ^b^	3.7 ± 1.2 ^b^	23.3 ± 1.5 ^c^	3.2 ± 0.7 ^b^	15.2 ± 1.3 ^c^
C18:3(n-3)	4.6 ± 0.9 ^a^	39.5 ± 2.2 ^b^	4.1 ± 1.1 ^b^	25.7 ± 2.2 ^c^	6.6 ± 1.3 ^b^	31.5 ± 2.3 ^c^
C20:0	traces	0.2 ± 0.0 ^a^	0.1 ± 0.0 ^a^	0.9 ± 0.1 ^a^	0.2 ± 0.0 ^a^	0.8 ± 0.1 ^a^
C20:1(n-9)	traces	0.3 ± 0.0 ^a^	0.2 ± 0.0 ^a^	1.2 ± 0.3 ^b^	0.1 ± 0.0 ^a^	0.6 ± 0.0 ^a^
C20:3(Δ11,14,17)	0.2 ± 0.0 ^a^	1.9 ± 0.9 ^b^	0.9 ± 0.1 ^a^	6 ± 1.3 ^c^	0.8 ± 0.2 ^a^	3.6 ± 1.1 ^b^
C20:4(Δ5,11,14,17)	0.3 ± 0.0 ^a^	2.4 ± 0.8 ^a^	0.4 ± 0.0	2.5 ± 0.9 ^b^	0.6 ± 0.1 ^a^	3 ± 0.9 ^b^
C22:0	traces	0.3 ± 0.0 ^a^	–	–	0.1 ± 0.0 ^a^	0.4 ± 0.0 ^a^
Total FAs	11.6 ± 1.1 ^b^	100	15.8 ± 1.2 ^a^	100	21.1 ± 1.6 ^b^	100
Σ_saturated_	3.4 ± 0.8 ^a^	29.4 ± 2.1 ^c^	5.2 ± 0.7 ^a^	32.9 ± 2.2 ^c^	7.5 ± 0.8 ^a^	35.7 ± 1.9 ^c^
Σ_unsaturated_	8.2 ± 0.9 ^b^	70.7 ± 1.9 ^c^	10.6 ± 1.1 ^b^	67.2 ± 2.4 ^b^	13.6 ± 1.1 ^c^	64.3 ± 2.1 ^c^
*k*		2.4		2		1.8
DBI		1.9		1.6		1.7
SDR		0.7		0.6		0.7
ODR		0.9		0.9		0.9
LDR		0.8		0.5		0.7

Note: “–”—acid not found; *—mean daily air temperature; Σsaturated—the sum of saturated FAs; Σunsaturated—the sum of unsaturated FAs; k—unsaturation coefficient; DBI—FA double-bond index. The table shows the average values from 3–6 biological replicates and their standard deviations. The significance of differences between the compared mean values was assessed using the *t*-test (*p* < 0.05), the normal distribution hypothesis was tested using the Shapiro–Wilk test. Different superscript letters indicate significant differences of analyzed parameters.

**Table 2 plants-10-00996-t002:** Seasonal changes in the FA composition of lipids in shoots of *Equisetum scirpoides* growing at the Pole of Cold.

	Summer Period (+17.8 °C *)		Autumn Period (+0.6 °C *)		Winter Period (−46.7 °C *)	
Fatty Acids						
	mg/g DW	%	mg/g DW	%	mg/g DW	%
C12:0	traces	0.3 ± 0.0 ^a^	traces	0.2 ± 0.0 ^a^	–	–
C14:0	0.2 ± 0.0 ^a^	1.3 ± 0.2 ^a^	0.1 ± 0.0 ^a^	0.8 ± 0.1 ^a^	0.1 ± 0.0 ^a^	0.7 ± 0.1 ^a^
C15:0	traces	0.3 ± 0.0 ^a^	0.1 ± 0.0 ^a^	0.3 ± 0.0 ^b^	–	–
C16:0	2.7 ± 0.9 ^a^	21.1 ± 1.2 ^c^	3.9 ± 0.9 ^a^	22.6 ± 1.6 ^c^	5.9 ± 1.1 ^b^	31.3 ± 2.3 ^c^
C16:1	0.1 ± 0.0 ^a^	1 ± 0.2 ^a^	0.2 ± 0.0 ^a^	1.4 ± 0.6 ^a^	traces	0.2 ± 0.0 ^a^
C16:2(n-6)	0.1 ± 0.0 ^a^	0.7 ± 0.1 ^a^	0.2 ± 0.0 ^c^	1.2 ± 0.5 ^a^	–	–
C16:3(Δ7,11,14)	0.3 ± 0.0 ^a^	2.6 ± 0.9 ^b^	1.0 ± 0.2 ^a^	5.7 ± 1.1 ^c^	1 ± 0.4 ^a^	5.2 ± 1.4 ^b^
C17:0	0.1 ± 0.0 ^a^	0.7 ± 0.1 ^a^	0.1 ± 0.0 ^b^	0.3 ± 0.0 ^a^	0.1 ± 0.0 ^a^	0.6 ± 0.2 ^a^
C18:0	0.7 ± 0.1 ^a^	5.2 ± 1.1 ^c^	0.3 ± 0.0 ^a^	1.8 ± 0.3 ^a^	0.3 ± 0.0 ^a^	1.8 ± 0.4 ^a^
C18:1(n-9)	0.8 ± 0.1 ^a^	6.1 ± 1.4 ^c^	0.6 ± 0.0 ^a^	3.7 ± 1.3 ^c^	0.9 ± 0.2 ^b^	4.8 ± 1.1 ^c^
C18:1(n-7)	0.1 ± 0.0 ^a^	0.6 ± 0.1 ^a^	0.1 ± 0.0 ^a^	0.3 ± 0.0 ^a^	0.1 ± 0.0 ^a^	0.3 ± 0.0 ^a^
C18:2 (Δ5,9)	0.7 ± 0.2 ^a^	5.8 ± 0.9 ^b^	0.5 ± 0.1 ^a^	3.1 ± 0.8 ^a^	0.5 ± 0.0 ^a^	2.4 ± 0.6 ^a^
C18:2(n-6)	2.0 ± 0.4 ^b^	16.1 ± 1.4 ^c^	2.3 ± 0.6 ^b^	13.1 ± 1.1 ^c^	3.6 ± 0.8 ^b^	18.8 ± 1.4 ^c^
C18:3(n-3)	4.0 ± 0.9 ^c^	31.4 ± 1.6 ^c^	6.5 ± 1.0 ^c^	37.5 ± 2.7 ^c^	5.9 ± 1.1 ^c^	31 ± 1.9 ^c^
C20:0	0.1 ± 0.0 ^b^	0.8 ± 0.1 ^a^	traces	0.2 ± 0.0 ^b^	0.1 ± 0.0 ^a^	0.5 ± 0.0 ^a^
C20:1(n-9)	traces	0.1 ± 0.0 ^a^	traces	0.1 ± 0.0 ^b^	–	–
C20:3(Δ11,14,17)	0.4 ± 0.1 ^a^	2.8 ± 0.7 ^b^	0.6 ± 0.1 ^a^	3.7 ± 1.1 ^c^	traces	0.2 ± 0.0 ^a^
C20:4(Δ5,11,14,17)	0.3 ± 0.0 ^b^	2.2 ± 0.6 ^b^	0.6 ± 0.1 ^a^	3.4 ± 1.3 ^b^	0.3 ± 0.0 ^a^	1.7 ± 0.3 ^a^
C22:0	0.1 ± 0.0 ^a^	0.8 ± 0.2 ^a^	0.1 ± 0.0 ^a^	0.5 ± 0.1 ^a^	0.1 ± 0.0 ^a^	0.5 ± 0.1 ^a^
Total FAs	12.6 ± 1.2 ^a^	100	17.2 ±1.4 ^c^	100	18.9 ± 1.2 ^b^	100
Σ_saturated_	3.8 ± 0.8 ^b^	30.5 ± 1.3 ^c^	4.6 ± 0.9 ^b^	26.7 ± 1.9 ^a^	6.7 ± 1.2 ^b^	35.4 ± 3.2 ^c^
Σ_unsaturated_	8.7 ± 0.9 ^b^	69.4 ± 1.9 ^c^	12.6 ± 1.3 ^c^	73.2 ± 2.2 ^c^	12.2 ± 2.3 ^c^	64.6 ± 3.4 ^c^
*k*		2.3		2.7		1.8
DBI		1.8		1.8		1.6
SDR		0.7		0.6		0.7
ODR		0.9		0.9		0.9
LDR		0.7		0.7		0.6

Note: “–”—acid not found; *—mean daily air temperature; Σsaturated—the sum of saturated FAs; Σunsaturated—the sum of unsaturated FAs; k—unsaturation coefficient; DBI—FA double-bond index. The table shows the average values from 3–6 biological replicates and their standard deviations. The significance of differences between the compared mean values was assessed using the *t*-test (*p* < 0.05), the normal distribution hypothesis was tested using the Shapiro–Wilk test. Different superscript letters indicate significant differences of analyzed parameters.

## Data Availability

All data are contained within the article.

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
