# Peer review of "Influence of Extremely Low Temperatures of the Pole of Cold on the Lipid and Fatty-Acid Composition of Aerial Parts of the Horsetail Family (Equisetaceae)"

_plants, 2021, doi:10.3390/plants10050996_

Round 1

Reviewer 1 Report

Please find my comments and suggestions in the pdf file. Thanks.

Author Response

We appreciate the reviewer’s comments. Changes are marked green in text.

The followings are our point-by-point responses:

  1. The introduction should introduce the different compounds measured in this study.

Response: Thank you for your recommendations. The section "Introduction" was complemented by information about lipids measured in this study. Changes are marked green in text.

  1. Statistics should be presented along with the results in all the figures.

Response: In fig. 3 and fig. 4 statistics data was added. (* - significant differences according to Student’s t-test.)

Tables 1 and 2 have been added «The significance of differences between the compared mean values was assessed using the t-test (P <0.05), the normal distribution hypothesis was tested using the Shapiro-Wilk test».

  1. Lines 168-169: “This section may be divided by sub-headings. It should provide a concise and precise description of the experimental results, their interpretation, as well as the experimental conclusions that can be drawn.” I do not know if one of the author indicated that, or if it was suggested by a previous reviewer. Anyway, I think it is a good suggestion to follow, the “results” section should be re-written. Conclusions should be present and clear.

Response:  Both you and the other reviewer commented on this section, so we are grateful to know that our current approach requires some rethinking. We significantly changed the section "Results", divided it by 2 sub-headings (2.1. Lipid content and 2.2. Fatty acid composition), as well as re-wrote the section "Discussion", adding explanations of data obtained, and added the section " Conclusions". Changes are marked green in text.

  1. Line 16: “an average daily air temperature in summer - 17.8°C”. According to the Figure 2, the average temperature in summer is a positive value, perhaps + 17.8°C? Please correct the text as needed.

Response: Corrections are made.

  1. Line 24: « linoleic C18:3» should be «linoleic C18:2”, I think.

Response: Yes, this is a typo. Thanks for noting it. Corrections are made.

  1. Figure 2: The dates of summer, autumn and winter should be mentioned in the legend of the Figure 2.

Response: Added seasonal periods to the Fig. 2

o “lippid content” should be replaced by “lipid content”

Response: Thanks for noting this typo. Corrections are made.

  1. Figure 3: The color legend should indicate if the temperatures are positive or negative.

Response:  Thank you for your recommendations. (green line – positive temperatures, blue line – negative temperatures) Fig. 2

-  Figure 4:

o “P...” refers for 3 different data sets, and is confusing: the full abbreviations should be written.

o The color legend should be completed.

Response: Corrected drawing according to your recommendations

  1. Line 332: the precise amount of C19:0 should be indicated.

Response: Thank you for adding the exact number of C19: 0

  1. The tables 1 and 2 should contain English characters only. If low amounts were detected,

the word “traces” could be used.

Response: Thanks for the advice, we fixed everything on "traces"

Reviewer 2 Report

Dear authors,

please, find out my suggestion within the comments in the document. In my opinion, you just need to fix some minor details, except two:

First, try to describe better the design of the experiment: how many samples did you have for chemical analyses of the biochemical compounds? How many samples of plant material did you collect, and how these samples are distributed among each chemical analysis?

Another thing I would like to suggest (and I have not emphasized this in the comment within the document): try to explain and better connect  the results and their context,  i.e. try to contextualize your findings. It is clear that FA contribute to plant hardening and stress tolerance, but I would like to see here in this paper the broader meaning of your results, and not just the pure chemical analyses. Could you maybe apply the same principles and conclusions in the field of breeding plants for stress tolerance? 

Author Response

We appreciate the reviewer’s comments.

Changes are marked green in text.

The followings are our point-by-point responses:

  1. First, try to describe better the design of the experiment: how many samples did you have for chemical analyses of the biochemical compounds? How many samples of plant material did you collect, and how these samples are distributed among each chemical analysis?

Response: Thank you for your recommendations. Added to the manuscript:

Equisetaceae plant samples were collected as an average biomass (2-5 g) consisting of several identical plants without a root system from a 1m2 plot in triplicate. The samples were immediately fixed in liquid nitrogen and transported in Dewar vessels to the laboratory. For biochemical studies, the samples of the aerial parts of horsetails fixed in liquid nitrogen were dried in the lyophilizer (VirTis, USA).

  1. Another thing I would like to suggest (and I have not emphasized this in the comment within the document): try to explain and better connect  the results and their context,  i.e. try to contextualize your findings. It is clear that FA contribute to plant hardening and stress tolerance, but I would like to see here in this paper the broader meaning of your results, and not just the pure chemical analyses. Could you maybe apply the same principles and conclusions in the field of breeding plants for stress tolerance? 

Response: Thank you your recommendations. We re-wrote the section "Discussion", significantly changed the section "Results", adding explanations of data obtained, and added the section " Conclusions". Changes are marked green in text.

Round 2

Reviewer 1 Report

The manuscript has been greatly improved. 

I only have minor comments:

  • Line 504: "docosahexaenoic C20: 6 acids" should be "docosahexaenoic C22: 6 acids", I think.
  • Line 438: "10 mcg" should be "10 mg", I guess.
  • In the tables 1 and 2, statistics should be indicated along each value. Indeed the reader needs to see, for each fatty acid, if the amounts are statistically significantly different between the 3 periods.

Author Response

We appreciate the reviewer’s comments.

The followings are our point-by-point responses:

  • Line 504: "docosahexaenoic C20: 6 acids" should be "docosahexaenoic C22: 6 acids", I think.

Response: Thank you for your recommendations. Corrected a typo in the manuscript.

  • Line 438: "10 mcg" should be "10 mg", I guess.

Response: We used 10 µg nonadecanoic acid (C19:0).

  • In the tables 1 and 2, statistics should be indicated along each value. Indeed the reader needs to see, for each fatty acid, if the amounts are statistically significantly different between the 3 periods.

Response: Thank you for your recommendations. Added to the manuscript in the Tables 1 and 2:

«Different superscript letters indicate significant differences of analyzed parameters».

We thank the anonymous reviewers for their comments.
